# A cocoon silk chemistry strategy to ultrathin N-doped carbon nanosheet with metal single-site catalysts

Youqi Zhu[1], Wenming Sun[2], Jun Luo [3], Wenxing Chen [1], Tai Cao[1], Lirong Zheng[4], Juncai Dong [4], Jian Zhang [1], Maolin Zhang[1], Yunhu Han[1], Chen Chen[1], Qing Peng[1], Dingsheng Wang [1] & Yadong Li[1]

Development of single-site catalysts supported by ultrathin two-dimensional (2D) porous matrix with ultrahigh surface area is highly desired but also challenging. Here we report a cocoon silk chemistry strategy to synthesize isolated metal single-site catalysts embedded in ultrathin 2D porous N-doped carbon nanosheets (M-ISA/CNS, M = Fe, Co, Ni). X-ray absorption fine structure analysis and spherical aberration correction electron microscopy demonstrate an atomic dispersion of metal atoms on N-doped carbon matrix. In particular, the Co-ISA/CNS exhibit ultrahigh specific surface area ($2105\,m^2\,g^{-1}$) and high activity for C–H bond activation in the direct catalytic oxidation of benzene to phenol with hydrogen peroxide at room temperature, while the Co species in the form of phthalocyanine and metal nanoparticle show a negligible activity. Density functional theory calculations discover that the generated O = Co = O center intermediates on the single Co sites are responsible for the high activity of benzene oxidation to phenol.

[1] Department of Chemistry, Tsinghua University, 100084 Beijing, China. [2] State Key Laboratory of Green Building Materials, China Building Materials Academy, 100041 Beijing, China. [3] Center for Electron Microscopy, Tianjin University of Technology, 300384 Tianjin, China. [4] Beijing Synchrotron Radiation Facility, Institute of High Energy Physics, Chinese Academy of Sciences, 100049 Beijing, China. These authors contributed equally: Youqi Zhu, Wenming Sun. Correspondence and requests for materials should be addressed to D.W. (email: wangdingsheng@mail.tsinghua.edu.cn) or to Y.L. (email: ydli@mail.tsinghua.edu.cn)

Single-site catalysts with isolated metal single-atomic sites supported by solid matrix are emerging as an exciting research frontier in catalysis[1–4]. With atomically metal dispersity, such single-site catalysts display several distinct advantages in the maximized atom efficiency, strengthened selectivity towards targeted product, improved intrinsic activity, and facilitative recyclability[5,6]. Indeed, single-site catalysts, including Au[7,8], Pt[9,10], Pd[11,12], Ru[13,14], Ir[15,16], Rh[17], Fe[18–20], Co[21,22], Ni[23–25], Mn[26], Mo[27], and W[28], are reported over past few years and exhibit interesting activity in CO oxidation, solar energy conversion, hydrogen production, hydrogenation, $CO_2$ reduction, and oxygen reduction reaction. To date, several synthetic protocols have been reported for fabricating single-site catalysts including photochemical reduction[2], co-precipitation[3], atomic layer deposition (ALD)[13], wet impregnation[29], and so on. Accordingly, the construction of support materials with rich anchoring sites and identical geometric structure is of fundamental importance for achieving desirable catalytic performance. Moreover, significantly enlarging surface area of support materials can expose the most active sites accessible to reactants, which has been considered as an effective route to boost the intrinsic activity of these heterogeneous supported single-site catalysts[30]. Various zeolites, metal–organic frameworks (MOFs), and well-defined porous carbon materials have been employed as possible support for single-site catalysts[31–34]. Therefore, establishing rational and general synthetic approaches to prepare single-site catalysts with controllable structure and ultrahigh surface area to take full advantage of their potential is highly desired but challenging.

The catalytic performances of single-site catalysts are highly dependent upon support materials[35,36]. Considerable researches have demonstrated that support materials can not only firmly stabilize isolated single metal sites but often play a key role in controlling the local geometric and electronic structures of catalytic sites, leading to excellent retentivity of individual site state under catalytic conditions and superb catalytic activity[1,14,37]. To date, the reported support materials generally consist of the oxides and carbon-based materials[38]. N-doped porous carbon materials have been widely employed for (i) rich anchoring sites, (ii) favorable mass adsorption and transportation facilitated by controllable porous structure with high surface area, and (iii) good thermal stability under harsh catalytic conditions[39,40]. Ultrathin 2D N-doped carbon nanosheet supported single-site catalysts stand for the ultimate low-end limit for active metal centers. Besides exposing the largest metal single-atomic sites on external surface, the ideal architecture can shorten diffusion pathway and provide sufficient interface area accessible to reactants, which has been well demonstrated in cease of graphene supported single-site catalysts[41,42]. Furthermore, the large planar area and atomic thickness of the such single-site catalysts make them a model platform to identify the accurate atomistic structure of active sites and the catalytic pathways in heterogeneous catalysis at molecular level.

The selective catalytic oxidation of aromatic inactive C–H bonds is one of the most active research topics but still remains a long-standing challenge[43–46]. Phenol is the most important industrial chemical due to its broad application in the fields of phenolic resin, pharmaceuticals, agrochemicals, etc[47,48]. However, current phenol production is still limited to the three-step cumene process, including propylation of benzene, autoxidation to cumene hydroperoxide, and Hock rearrangement[49–52]. These reactions are conducted under elevated temperature, high pressure, and strongly acidic conditions, which result a low overall yield of phenol (<5%) and an equimolar amount of acetone byproduct[50,51]. Thus, one-step direct catalytic oxidation of benzene to phenol with hydrogen peroxide over heterogeneous

catalysts has remained a focal subject of the fundamental and applied research[4,18,53–55]. Bal and co-workers have devoted great effort in such catalytic system based on novel heterogeneous catalysts including zeolite and $CuCr_2O_4$ spinel[43–45,48,56]. Despite considerable achievements, the direct catalytic oxidation of benzene at room temperature is still very difficult due to the notoriously low reactivity of the aromatic C−H bonds[57,58]. Development of more efficient catalysts has been the most urgent task. Given that the achieved high catalytic performance in most single-site catalysts, we expect their preparation will provide exciting opportunity in direct catalytic oxidation of benzene to phenol.

Here we report a cocoon silk chemistry strategy to prepare isolated metal single-atomic site catalysts embedded in ultrathin 2D porous N-doped carbon nanosheets (M-ISA/CNS, M = Fe, Co, Ni). Their atomistic structures were revealed by spherical aberration correction electron microscopy and X-ray absorption fine structure analysis. The as-synthesized Co-ISA/CNS exhibited a high nitrogen content (9.2 wt %) and large specific surface area (2105 $m^2 g^{-1}$). As a proof of concept, the Co-ISA/CNS catalyst showed high activity for C−H bond activation with 68% conversion in the direct catalytic oxidation of benzene to phenol with hydrogen peroxide at room temperature. The density functional theory (DFT) calculations were further performed to investigate the reaction mechanisms of benzene oxidation over the Co-ISA/CNS catalyst. Furthermore, the synthetic strategy demonstrates efficient and general for preparing single-site catalysts, providing opportunities for single-site heterogeneous catalytic applications in atomic scale.

## Results

**Concept and procedure of cocoon silk chemistry strategy.** Our cocoon silk chemistry synthetic strategy to Co-ISA/CNS consists of four steps: (i) cocoons degumming; (ii) silk fibroin regeneration; (iii) salts-assisted pyrolysis; and (iv) acid etching (Fig. 1a). This synthesis is based on the utilization of amino-group-rich chemistry and lamella-like sheet structure of silk fibroin to anchor single cobalt atom and simultaneously construct 2D morphology. Silk fibroin was first extracted from *Bombyx mori* cocoons through a modified degumming procedure[59]. The degummed silk fibroin was then dissolved in a concentrated aqueous solution of the cobalt and zinc salt. Driven by self-assembly of the hydrophobic and hydrophilic blocks, the silk fibroin was subsequently restructured into regenerated silk fibroin (Supplementary Fig. 1). The traditional all-aqueous extraction could convert natural silk fiber into lamella-like layer structure with β-sheet crystal contents. The regenerated silk fibroin exhibits a geometrically sheet-like 2D morphology (Supplementary Fig. 2) with a rich chemistry and high nitrogen content of 20.47 wt% (Supplementary Fig. 3). Concurrently, a large amount of cobalt and zinc ions were intercalated into layer space or absorbed on layer surface of regenerated silk fibroin, and tightly anchored into the unique precursor framework by strong interaction with amino groups. The excess metal salts not only promote dissolution of natural silk but also provide a hypersaline conditions and act as effective activation agents or porogens in pyrolysis process. Following a pyrolysis at 900 °C, the layered biopolymer film was in situ converted into 2D porous N-doped carbon nanosheets, meanwhile the rich amino groups were degraded to N species inserting into carbon skeleton to prevent the aggregation and migration of previously incorporated cobalt ions, resulting isolated single cobalt atoms. Finally, Co-ISA/CNS was obtained through acid leaching to remove soluble salts and generated cobalt nanoparticles.

The X-ray powder diffractometer (XRD) pattern (Supplementary Fig. 4a) of Co-ISA/CNS exhibits a broad graphitic (002) peak

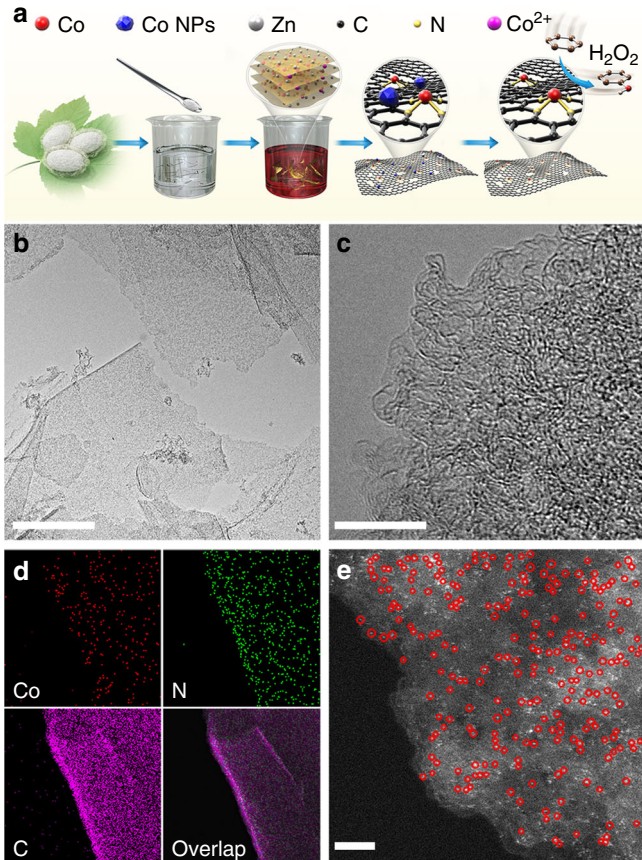

**Fig. 1** Synthesis and characterizations of Co-ISA/CNS catalyst. **a** Illustration of the proposed formation mechanisms. **b** TEM image. Scale bar, 500 nm. **c** HRTEM image. Scale bar, 10 nm. **d** EDX elemental mapping. **e** Aberration-corrected HAADF-STEM image. Scale bar, 2 nm

at around $2\theta =$ about 26°, which belongs to the hexagonal conjugated carbon structure. There were no characteristic peaks of metal Co crystals, demonstrating that the isolated large metal nanoparticles have been removed completely by acid etching. Low-magnification field emission scanning electron microscopy (FESEM) and transmission electron microscopy (TEM) images (Supplementary Fig. 4b, e) show that the Co-ISA/CNS presents a geometrically sheet-like 2D structure with uniform morphology and large planar area, and no other carbon configurations are observed. Supplementary Fig. 4c, d further display a highly flexible and gauze-like morphology of Co-ISA/CNS. Figure 1b and Supplementary Fig. 4f clearly reveals a transparent feature, which belongs to the fundamental characteristic in common of this type of ultrathin 2D graphene-like nanostructure. Additionally, a continuous 2D porous network is clearly observed from the TEM images. The high-resolution TEM (HRTEM) image (Fig. 1c) reveals some irregular fringes, corresponding few graphene layers. An amount of disordered pores with diameter of 3–4 nm can also be found, and no cobalt particles are present. energy-dispersive X-ray spectroscopy (EDX) mapping analysis (Fig. 1d) suggests that Co, N, and C are homogeneously distributed on the whole carbon nanosheet. The Co content was determined to be about 0.6 wt % by inductively coupled plasma optical emission spectrometry (ICP-OES) analysis (Supplementary Table 1). The atomically dispersed single Co atoms can be clearly distinguished as the brighter spots on N-doped carbon nanosheet from aberration-corrected high-angle annular dark-field scanning transmission electron microscopy (HAADF-STEM) image (Fig. 1e). Several Co single atoms were circled in red for close observation.

X-ray photoelectron spectroscopy (XPS) exhibits three obvious characteristic peaks of C, N, and O elements as well as a clear high-resolution Co 2p spectrum (Supplementary Fig. 5). The deconvoluted core level spectrum of the N 1 s region (Supplementary Fig. 5b) consists of two strong peaks at 400.6 and 398.4 eV, corresponding to pyrrolic-N and pyridinic-N, respectively, with N content of 9.2 wt %. Raman spectrum (Supplementary Fig. 6) presents two obvious signals at 1335 and 1593 cm$^{-1}$, which are related to the D and G bands, respectively. The observed G-band suggests a typical sp$^2$-hybridized carbon network, while the broad D-band with relatively high intensity reveals highly defective structure of Co-ISA/CNS. Additionally, the graphene-like Co-ISA/CNS exhibits high specific surface area with porous structure (Supplementary Fig. 7). The 2D porous nature of Co-ISA/CNS caused a large Brunauer–Emmett–Teller (BET) specific surface area of 2105 m$^2$ g$^{-1}$, which was comparable to the theoretical value of monolayer graphene (2630 m$^2$ g$^{-1}$). The dominant pore sizes distribution is calculated to be about 3.4 nm with a high total pore volume of 1.7 cm$^3$ g$^{-1}$.

Element-selective X-ray absorption fine structure (XAFS) measurements at Co K-edge were performed to further investigate the chemical state and coordination environment of Co species at atomic level. The results show that the Co species are present in the form of isolated dispersed single Co atoms, and stabilized by nitrogen in CoN$_4$ atomic structure. Figure 2a shows the normalized X-ray absorption near edge structure (XANES) curves of Co K-edge for Co-ISA/CNS along with Co foil, CoO, and Co$_3$O$_4$ references. Being significantly different from those of the references, the Co-ISA/CNS presented relatively smooth XANES profile. The position of absorption edge of Co-ISA/CNS located between those of the CoO and Co$_3$O$_4$ clearly demonstrates the unique electronic structure with positive charges in a valence state of Co$^{\delta+}$ ($2 < \delta < 3$), mainly due to the nitrogen doping. Moreover, the pre-edge peak at 7709.6 eV is caused by a $1s$-$4p_z$ transition, indexing the fingerprint of Co–X$_4$ coordination (here X = N). As shown in Fig. 2b, the Co-ISA/CNS exhibits a main peak at 1.41 Å that is shorter than Co-O peak at 1.71 Å in CoO and 1.51 Å in Co$_3$O$_4$, and thus can be ascribed to the backscattering between Co and light atoms, corresponding to the Co−N/C first coordination shell. By contrast, the main peak for Co foil is observed at 2.20 Å, which can be ascribed to the metallic Co−Co coordination. The metallic Co−Co peak for the Co-ISA/CNS is negligible, and no other high shell peaks are detected. This result suggests that the Co atoms in Co-ISA/CNS are atomically dispersed. Additionally, the wavelet transforms (WT) of Co K-edge extended X-ray absorption fine structure (EXAFS) oscillations also well demonstrate the atomic dispersion of Co species in Co-ISA/CNS. As shown in Fig. 2c, the WT contour plots of Co-ISA/CNS present only one intensity maximum at 4.0 Å$^{-1}$ that can be assigned to the Co−N/C coordination and no intensity maximum related to Co−Co coordination can be observed, compared with that of Co foil, CoO, and Co$_3$O$_4$ references. The coordination configuration of Co atom for Co-ISA/CNS was investigated by quantitative EXAFS curve fitting analysis (Fig. 2d, e). The best-fitting analyses clearly present a main peak originating from Co−N first shell coordination, which is evidently different from that of Co foil (Supplementary Fig. 8). The calculated coordination number of center Co atom is about 4 and the mean bond length of Co-ISA/CNS is 1.98 Å (Supplementary Table 2). So based on the experimental spectra results and the well-matched EXAFS fitting, the local atomic structure model of Co-ISA/CNS can be identified as shown by the inset in Fig. 2e, where one Co atom is coordinated by four N atoms (CoN$_4$) confined in the carbon matrix. However, it is necessary to point out that the pyrolytic carbonaceous materials converted from metal-contained

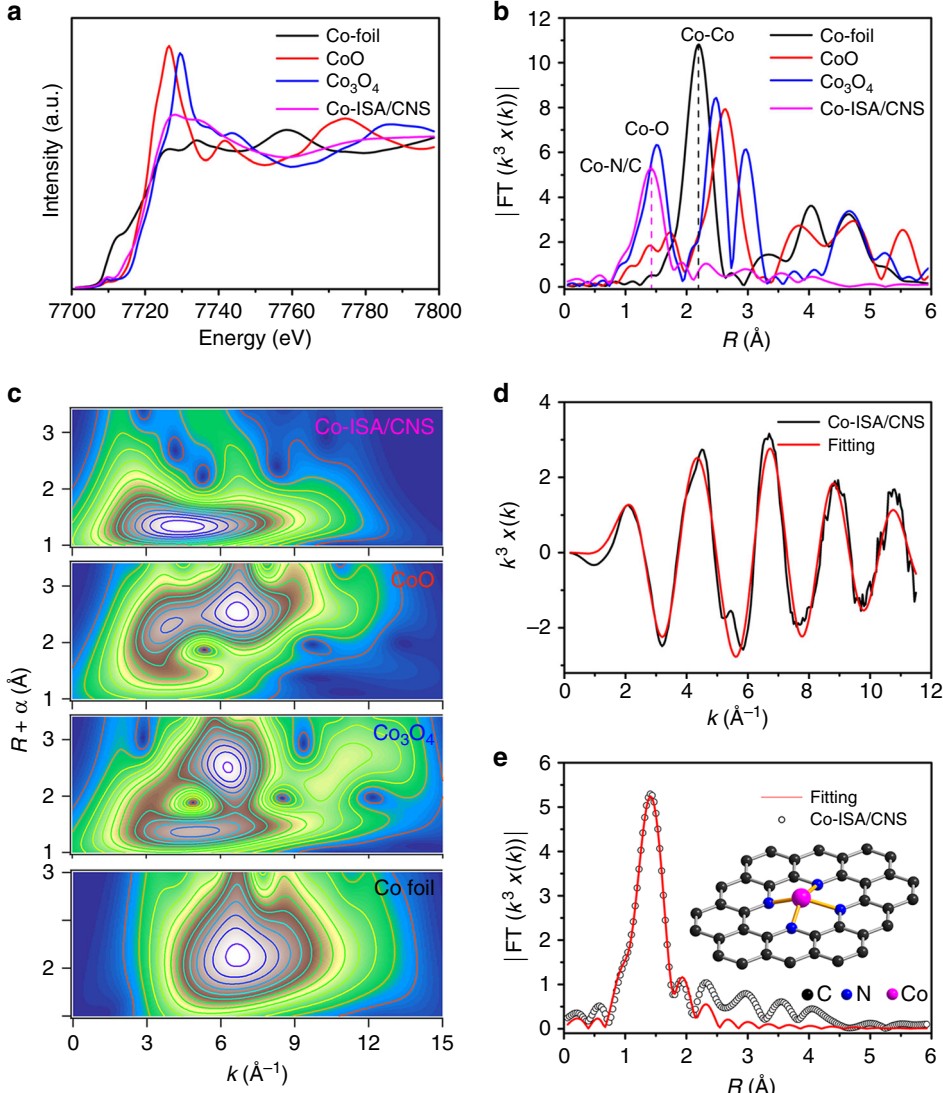

**Fig. 2** Synchrotron XAFS measurement of Co-ISA/CNS catalyst. **a** Co K-edge XANES spectra of Co-ISA/CNS catalyst and reference samples. **b** Fourier transformed (FT) $k^3$-weighted $\chi(k)$-function of the EXAFS spectra for Co K-edge. **c** Wavelet transforms for the $k^3$-weighted EXAFS signals. **d**, **e** Corresponding EXAFS fitting curves at $k$ and $R$ space, respectively, inset showing the schematic model

precursors may usually deliver various metal-nitrogen moieties ($MN_x$, $x = 0$, 1, 2, 3, and 4) in carbon lattices caused by the disorder, edge, and in-plane atomic vacancy during the thermal treating process at high temperature. Furthermore it is difficult to completely exclude the existence of other metal-nitrogen moieties based on the current experimental characterization technology. So we carried out the DFT calculations on formation energies of various possible structures to examine the stability of $CoN_x$ ($x = 0$, 1, 2, 3, and 4) and to access whether there are $CoN_3$, $CoN_2$, $CoN$, or $CoC$ existing in the carbon matrix. The DFT calculations were conducted following a modified procedure based on that reported in Deng's work[41], and the detailed methods and results are summarized in Supplementary Fig. 9. After comparing the formation energies of different structures, it is evidently found that the $CoN_4$ moiety is the most stable compound among $MN_x$ structures. Therefore we propose that the $CoN_4$ moieties are the most possible atomic structures for Co species and serve as the dominant active sites in Co-ISA/CNS catalyst.

By skillfull taking advantage of amino-group-rich chemistry and lamella-like sheet structure of silk fibroin, the Co-ISA/CNS catalysts are successfully synthesized for the first time. It is found

that the regenerated silk fibroin precursor plays a crucial role in the formation of the sheet-like 2D structure. When replacing silk fibroin precursor by melamine, the identical treatments results into a hybrid of tube-like 1D carbon microstructure and Co nanoparticles (Supplementary Fig.10), in which the Co nanoparticles are tightly encapsulated by multiple carbon layers and could survive in acid leaching process. However, for regenerated silk fibroin precursor, their lamella-like layer 2D backbone could be maintained in pyrolysis process. The reduced Co nanoparticles are anchored on surface of converted carbon layer and could be removed by acid leaching.

Furthermore, we also found that the cocoon silk chemistry strategy proves versatile in synthesis of Co single-site catalysts with tailorable central-metal atomic local structures. Our previously published researches have demonstrated that a certain amount of coordination bonds between N and metal atoms in pyrolytic carbon matrix can be broken up with pyrolysis temperature increasing[60]. Here we synthesized different samples at much higher pyrolysis temperature. Microstructure characterizations and XAFS analysis (Supplementary Fig. 11, 12, 13, 14) suggest that both the samples exhibit a similar morphology of the

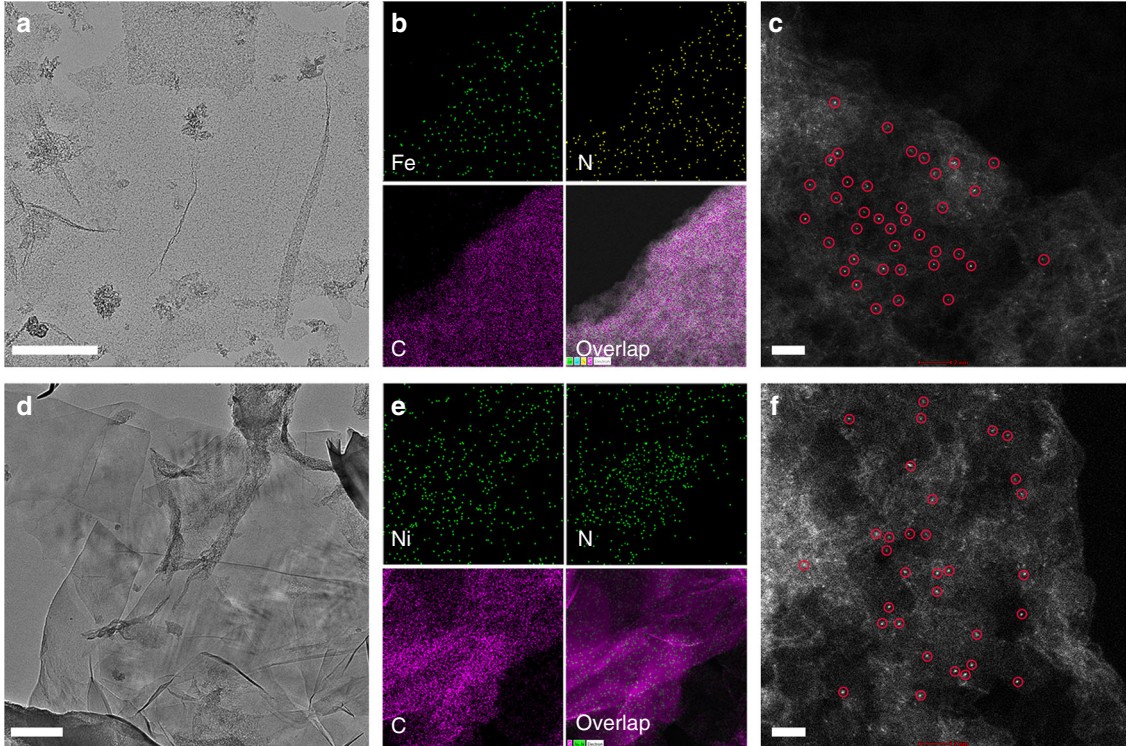

**Fig. 3** Microstructure and morphology characterizations of Fe-ISA/CNS and Ni-ISA/CNS. **a–d** Typical TEM images. Scale bar, 500 and 200 nm. **b–e** Corresponding EDX elemental mapping. **c–f** Aberration-corrected HAADF-STEM images. Scale bar, 2 nm

ultrathin 2D porous N-doped carbon nanosheets with atomically dispersed Co sites. The dominant reactive sites for Co single-site catalysts obtained at 950 and 1000 °C are identified as CoN$_3$ and CoN$_2$, designated as Co-ISA/CNS-3 and Co-ISA/CNS-2, respectively.

Additionally, the present cocoon silk chemistry strategy is also versatile in synthesis of porous carbon materials with ultrahigh specific surface area. In general, high surface areas and large pore volumes are highly required for heterogeneous catalysts. Chemical activation followed by carbonization process has been extensively employed to achieve such materials using different chemical activating reagents, typically KOH, H$_3$PO$_4$, or ZnCl$_2$, etc[61,62]. The ZnCl$_2$ activation approach can enjoy many advantages of low cost, and minor erosion. During carbonization process, the ZnCl$_2$ salt shows strong interaction or coordination with the carbon π-electron organism. At elevated temperature beyond the boiling point of ZnCl$_2$ (756 °C)[62], it can be evaporated away completely, where etching effect toward C−C bonds takes place, leaving numerous carbon atom vacancy and further evolving into abundant nanopores. So it can be concluded that the quantity of ZnCl$_2$ salt used in synthesis has crucial impacts upon the as-synthesized carbon texture. Supplementary Fig. 15, 16 show the microstructure characterizations of the Co-ISA/CNS catalysts synthesized with reduced amount of with ZnCl$_2$ salt (5.5 mmol) and in the absence of ZnCl$_2$ salt, respectively. Although the typical ultrathin 2D morphology can be well maintained, both the samples exhibit much less nanopores. Especially, the Co-ISA/CNS catalysts synthesized without ZnCl$_2$ salt show a relatively smooth surface, indicative of disappearing of in-plane porous structure. The BET surface areas of Co-ISA/CNS catalysts synthesized with 5.5 mmol and without ZnCl$_2$ salt are measured to be 1781 and 1061 m$^2$ g$^{-1}$ (Supplementary Fig. 17).

Considering the capability of a series of metal salt to dissolve silk fibroin in aqueous solution, we extended this strategy to other metal precursors and successfully synthesized a family of analogous M-ISA/CNS (M = Fe, Ni) under identical experimental conditions. As shown in Fig. 3a–d and Supplementary Fig. 18, 19, they all exhibit a similar 2D porous sheet-like morphology with a highly flexible and transparent feature. Figure 3b, e reveal that the corresponding Fe and Ni elements are homogeneously distributed on the whole ultrathin 2D porous N-doped carbon nanosheets, respectively. Aberration-corrected HAADF-STEM images (Fig. 3c–f) and XAFS (Supplementary Fig. 20, 21) further demonstrate the feature of isolated metal single-atomic sites anchored carbon matrix. Therefore, this silk fibroin strategy is very effective and versatile for fabrication of isolated metal single-atomic sites supported on 2D porous carbon matrix, which may open up more opportunities for single-site catalysts in broad applications.

**Catalytic performance for C−H bond activation.** Our catalytic measurements show that the as-synthesized Co-ISA/CNS catalysts could exhibit high activity for the direct catalytic oxidation of benzene to phenol with hydrogen peroxide at room temperature (25 °C). Figure 4a presents a comparison of benzene oxidation catalyzed by Co-ISA/CNS, cobalt phthalocyanine (CoPc), Co nanoparticles/N-doped carbon (Co-NPs), and N-doped carbon (NC). When the reaction is conducted over pure NC (Supplementary Fig. 22), only 0.3% benzene conversion was achieved. Even if loading some cobalt species on NC support in nanoparticles form (Supplementary Fig. 23), only 0.8% benzene conversion is detected. However, when constructing Co-ISA/CNS catalysts, the benzene conversion is promoted to be 68% with 61% phenol yields. To the best of our knowledge, the presented catalytic performance is superior to the most results of heterogeneous catalysts reported in literature[53,54,57,63,64]. For comparison, the homogeneous CoPc catalysts exhibit benzene conversion of 1%. The Co-ISA/CNS catalysts show a turnover frequency of 150 h$^{-1}$ for benzene conversion within the initial 1 h (Supplementary Fig. 24). Moreover, the phenol selectivity over

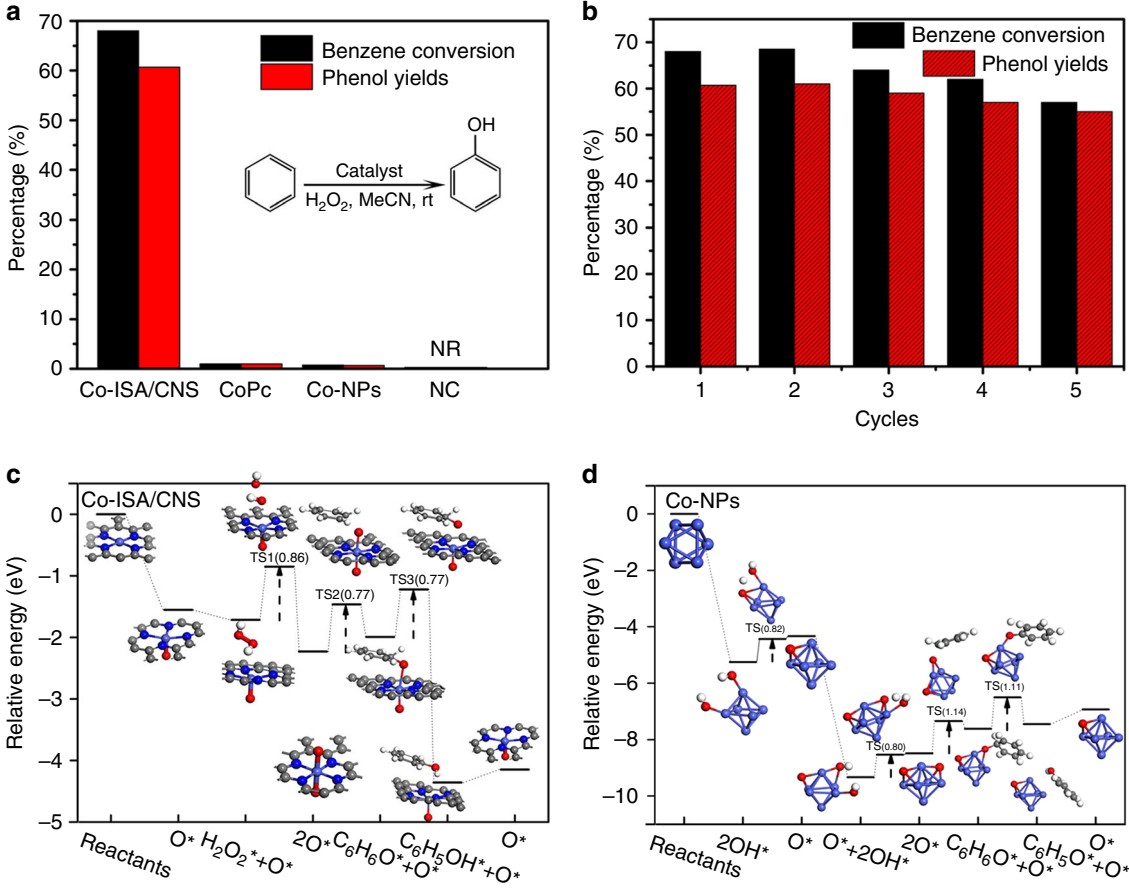

**Fig. 4** Catalytic performance and proposed mechanism. **a** A comparison of benzene oxidation catalyzed by the Co-ISA/CNS catalyst, CoPc, Co-NPs, and NC. **b** Recycle performance of Co-ISA/CNS catalyst. **c, d** Proposed mechanism of $H_2O_2$ activation and benzene oxidation by DFT calculations over Co-ISA/CNS catalyst and Co-NPs, respectively

the Co-ISA/CNS catalysts reaches 97% within the initial 1 h, but decreases progressively with increasing the reaction time and maintains at a value of 90% at the reaction time prolonged to 96 h. The side-product is detected to be hydroquinone, which is over-oxidized from benzene through multiple oxidations during the liquid phase reaction process and similar to that reported in literatures[63,64]. Additionally, as shown in Supplementary Fig. 25 that the hydrogen peroxide efficiency over the Co-ISA/CNS catalysts increases from 1.1% to 4.7% in the catalytic reaction process, which is related to the $H_2O_2$ decomposition rate and the benzene oxidation rate. It is therefore evident that the Co-ISA/CNS can serve as a new class of fascinating catalyst for efficiently catalytic oxidation of benzene to phenol with hydrogen peroxide at room temperature.

Bal's work demonstrated that several reaction parameters, such as reaction temperature, composition of reaction medium, amount of oxidant, and type of catalyst, may have important effect on the benzene oxidation reaction in liquid phase over heterogeneous catalysts[56]. Supplementary Fig. 26 suggests that the benzene oxidation reaction over the Co-ISA/CNS catalyst is highly dependent on reaction temperature. When the reaction carried out at 0 °C, the Co-ISA/CNS catalyst can deliver 14.3% benzene conversion after 24 h, which is higher than that 8.3% of Fe-based catalyst under the identical conditions[18], indicative of excellent catalytic performance. With increasing reaction temperature, both benzene conversion and phenol yields increased, but phenol selectivity decreased slightly. Supplementary Fig. 27 shows the effect of the amount of $H_2O_2$ in benzene oxidation over

the Co-ISA/CNS catalyst. It can be observed that increasing the amount of $H_2O_2$ results in higher conversion of benzene. However, when the amount of $H_2O_2$ was further increased to 7 ml, the benzene conversion exhibited a little decrease. Supplementary Fig. 28 shows that solvents can also play a very critical role in benzene oxidation over the Co-ISA/CNS catalyst. Among the solvents, the acetonitrile proves to be the best reaction medium mainly due to its excellent capability to well dissolve substrate benzene and offer a highly dispersed system for solid catalyst. When reacted in the highly hydrophobic n-octane, the Co-ISA/CNS catalyst showed the worse activity. In order to insight into the structure-activity relationship of the as-synthesized Co single-site catalyst towards the benzene oxidation reaction, we also measured the catalytic performance of another two samples with different N coordination numbers under the same conditions. As revealed in Supplementary Fig. 29, the Co-ISA/CNS catalyst with $CoN_4$ structure achieved the highest catalytic activity in comparison with Co-ISA/CNS-3 ($CoN_3$) and Co-ISA/CNS-2 ($CoN_2$). The resultant difference in catalytic activity is speculated to be related to their structural stability in practice reaction conditions. In case of the most stable $CoN_4$ structure, the sufficient reactive centers can be effectively build up and maintain the longest lifetime, which could endow the highest catalytic activity. In transition-metal-catalyzed C−H bond activation, the generated metal sites play a key role in achieving desirable catalytic activity. To clarify the effect of metal sites toward catalytic activity, Fe-ISA/CNS and Ni-ISA/CNS catalysts were also used for the benzene oxidation reaction. As shown in

Supplementary Fig. 30, Fe-ISA/CNS and Ni-ISA/CNS catalysts are found to be much less active than Co-ISA/CNS catalyst, indicative of the positive role of Co sites.

In addition to the remarkable catalytic activity, the high recycling stability and reusability of the Co-ISA/CNS catalyst are also an advantage worth addressing. As shown in Fig. 4b, the Co-ISA/CNS catalyst could be easily recycled by centrifugation from the solution and reused at least five times without obvious loss of activity and selectivity. The microstructure and morphology characterizations (Supplementary Fig. 31), as well as EXAFS spectrum (Supplementary Fig. 32) of recycled Co-ISA/CNS catalyst show that their structural integrity and stability are still maintained unchanged. The atomic dispersion of Co species is well preserved after repetitive runs, and no aggregation is observed, unambiguously demonstrating the excellent stability and recycling ability of the Co-ISA/CNS catalyst.

To get more insight into the high catalytic activity of the Co-ISA/CNS catalysts, we performed comparative DFT calculations on the reaction mechanisms of benzene oxidation over Co-ISA/CNS, CoPc, and Co nanoparticles ($Co_6$ cluster). Figure 4c shows the free energy profile and reaction pathway of benzene oxidation on the Co-ISA/CNS catalyst. Generally, two key steps can be abstracted in the reaction pathway including the initial formation of activated oxygen species by the $H_2O_2$ decomposition and the subsequent oxidation of benzene to phenol by the activated oxygen species. The catalytic reaction process originates from the adsorption of oxidant $H_2O_2$ molecule on the single Co atomic site. Three adsorption models were proposed, and the corresponding adsorption energy is shown in Supplementary Fig. 33. Interestingly, compared with model $a$ and $b$, here it shows that model $c$ exhibited the lowest total energy ($E_{rel}$), indicating the most stable structure. The result suggests that the $H_2O_2$ adsorption on the single Co atomic site followed the pathway shown in mode $c$, i.e., an oxidant $H_2O_2$ molecule was initially adopted on the single Co atomic site and then easily dissociated by forming a $Co=O$ intermediate and releasing one water molecule. After that, another $H_2O_2$ molecule was continuously adopted and dissociated on the other side of the Co atomic site, and then an $O=Co=O$ center was established. This process is energetically favorable due to preceding an exothermic step and the moderate energy barrier of 0.86 eV. The generated $O=Co=O$ center could present activated oxygen species to adopt the benzene molecule via the C–O bonding by overcoming an energy barrier of 0.77 eV. The adsorbed benzene was subsequently converted to phenol via an H atom transfer from adjacent C to O with an energy barrier of 0.77 eV. After releasing the phenol from active site, the $Co=O$ site was regenerated in the end of one reaction cycle. The energy barriers to overcome for the three transition states (TS1, TS2, and TS3) are relative low, and the whole reaction pathway was calculated to be exothermic, which are favorable for low temperature reactions.

Previous computational and experimental studies have evidently revealed that the high-valent metal–oxo species generally serve as reactive intermediates in the catalytic oxidation of organic substrates[65]. In particular, for the cobalt-mediated C–H oxidation reactions, the reactive intermediates are frequently implicated to cobalt(IV)–oxo species[66,67]. However, it is difficult to identify the proposed cobalt(IV)–oxo intermediates under catalytic conditions, and only extremely rare cases have achieved some direct spectroscopic characterizations. To investigate both coordination environment and oxidation state of cobalt sites in the used Co-ISA/CNS catalysts, the cobalt K-edge XAFS measurements were first performed. As shown in Supplementary Fig. 32, the edge position was found to locate at 7721.64 eV at $A = 0.5$ edge-jump, which is ~0.84 eV blue-shifted from the reported Co(III)–peroxo complexes[65]. Although the observed

energy shift is smaller than that valence change from Co(III) to Co(IV), it is still indicative of highly oxidized cobalt sites. Additionally, the used Co-ISA/CNS catalysts delivered higher white line intensity than that of the fresh Co-ISA/CNS catalysts, demonstrating the more positive average oxidation state of cobalt sites in the used Co-ISA/CNS catalysts[28]. The EXAFS curves of the used Co-ISA/CNS catalysts were best modeled with cobalt sites in an about five number N/O coordination environment. These XAFS results strongly indicate that the cobalt active sites in the Co-ISA/CNS catalysts are generated as a square pyramidal $Co^{IV}$–O-like species[65]. This conclusion is also well evidenced by the electron paramagnetic resonance (EPR) spectroscopy and XPS data analysis of the Co-ISA/CNS samples before and after the $H_2O_2$ treatment. As shown in Supplementary Fig. 34a, EPR spectrum of the Co-ISA/CNS catalyst with $H_2O_2$ treatment shows signals at $g = 6.6$ and 2.01, indicating the spin-quartet ($S = 3/2$) ground state[65,68]. The enhanced EPR signals could be ascribed to the probable formation of $Co=O/O=Co=O$, leading to Co 3d mixing with O 2p. The interaction between Co atomic sites and activated oxygen species will enhance the unoccupied state of Co species because of the electronegativity of cobalt-bonding O atoms. Supplementary Fig. 34b presents the binding energy variety of Co 2p from the fresh, $H_2O_2$-treated and spent Co-ISA/CNS catalysts based on XPS analysis. All spectra exhibit one pair of peaks relative to Co $2p_{3/2}$ and Co $2p_{1/2}$. The Co $2p_{3/2}$ of the fresh Co-ISA/CNS catalysts occurs at 780.8 eV, which is slightly higher than that of $Co^{II}$ but smaller than $Co^{III}$ chemical state. Interestingly, the Co $2p_{3/2}$ displays a positive energy shift of ~1.1 eV after $H_2O_2$ treatment, which is higher than the value of $Co^{III}$ chemical state in literature[69]. Thus, our current findings should support the involvement of the $O=Co=O$ center as the most plausible reactive intermediate in this cobalt-catalyzed oxidation reaction.

For Co nanoparticles, the adopted $H_2O_2$ was dissociated into two hydroxyls adsorbing on two adjacent Co atoms (Fig. 4d). After that, one hydroxyl shifted to the other for interaction by releasing one water molecule, this reaction relates a transition state and needs to overcome an energy barrier of 0.82 eV (Supplementary Fig. 35). Undergoing additional elementary reactions, the $O=Co=O$ center on Co nanoparticles was established. However, the subsequent oxidation reaction toward benzene ring is highly endothermic. Additionally, much higher energy barrier over 1.0 eV is required for the last two steps. These results make the benzene oxidation hard to take place at low temperature over Co nanoparticles. Supplementary Fig. 36 shows that benzene oxidation over CoPc undergoes a similar reaction pathway to that on Co-ISA/CNS, but the $H_2O_2$ molecule was initially dissociated on Co site forming an OH-Co-OH intermediate. In the $H_2O_2$-$H_2O$-$CH_3CN$ system, the OH-Co-OH intermediate was easily destroyed for the observation of severely bubbling in the practical oxidation reaction, indicative of a strong tendency to self-decomposition of hydrogen peroxide rather than contributing to hydroxylation reaction, which may break off the following processes and result into the deactivation of Co site. Therefore, the high activity for benzene oxidation to phenol over Co-ISA/CNS catalyst could be ascribed to the intrinsic coordination environment and the atomic dispersion of Co species in Co-ISA/CNS, which provides a novelty in reaction pathway much more superior to that over traditional heterogeneous catalysts.

## Discussion

In summary, we report a general cocoon silk chemistry strategy to synthesize metal single-site catalysts embedded in 2D porous N-doped carbon nanosheets (M-ISA/CNS, M = Fe, Co, Ni). This synthesis is based on the utilization of amino-group-rich

chemistry and lamella-like sheet structure of silk fibroin. With the assistance of metal salts in solution and pyrolysis process, silk fibroin is transformed into 2D porous N-doped carbon nanosheets for anchoring metal single-atomic sites. The X-ray absorption fine structure analysis and spherical aberration correction electron microscopy demonstrate an atomic dispersion of metal species on carbon matrix. Our experimental measurements show a surprising transformation in activity for direct catalytic oxidation of benzene to phenol at room temperature from inert state to high activity over Co species in the form of metal nanoparticle and isolated single metal atomic sites. DFT calculations discover that the high activity could attributed to the generated O=Co=O center intermediates on single Co sites. Our results offer a reliable approach for configuring metal single-atomic site catalysts and more opportunities for heterogeneous catalytic applications beyond the C-H bond activation in atomic scale.

## Methods

**Synthesis of Co-ISA/CNS.** All silk fibroin used in this work were first extracted from *Bombyx mori* cocoons through a modified degumming procedure reported elsewhere[70]. 6 g of cocoons were boiled for 3 h in a 0.2 M $Na_2CO_3$ aqueous solution. The resulting silk fibroin suspension was then rinsed thoroughly with water and purified by centrifugation at 14,000 rpm over two 10-min-long periods to remove the glue-like sericin proteins, and then the degummed silk fibroin was obtained. The degummed silk fibroin was subsequently dissolved in 100 mL aqueous solution of 0.25 M $CoCl_2$ and 0.11 M $ZnCl_2$ under continuously stirring at 80 °C for 5 h, yielding a regenerated silk fibroin solution. Subsequently, the solution was evaporated and transferred into a solid mixture. The pyrolysis process was conducted in a tubular furnace under an Ar atmosphere with a heating rate of 3 °C $min^{-1}$ up to 900 °C for 1 h. After allowing the system naturally cooled down to room temperature, the resulting dark solid powder was transferred into a 1 M HCl solution and refluxed at 120 °C for 12 h to remove soluble salts and generated cobalt nanoparticles. Finally, the Co-ISA/CNS was collected by centrifugation and then dried in oven overnight after washing by distilled water and anhydrous ethanol for several times.

**Catalyst evaluation.** The catalytic performances were evaluated through a modified process according to the previously published work and the recent publication from our group[4,18]. Unless otherwise noted, the catalytic benzene oxidation evaluation was carried out as following: the reaction was conducted in a 25 ml sealed glass reactor with 50 mg catalyst, 0.4 ml benzene, 6 ml $H_2O_2$ (30%), and 3 ml $CH_3CN$ at room temperature (25 °C). After the reaction, the mixture was extracted with ethyl acetate, while the solid catalyst was recovered by centrifugation. The products were analyzed with GC−MS and GC with n-hexadecane as an internal standard. General procedure for recyclability test: After each reaction, the catalyst was separated from the reaction system by centrifugal separator (HERMLE Z32 HK) at a 18,400× *g* spin rate for 10 min. Then the collected catalyst was washed by water and ethanol three times, and then dried at 80 °C for 12 h. The obtained powder was submitted to the next batch of catalytic reaction. The yield of phenol was calculated as (mole of formed phenol)/(mole of initial benzene) × 100%. The selectivity of phenol was calculated as (mole of formed phenol)/(mole of formed phenol + mole of formd benzoquinone) × 100%.

## Data availability

The data supporting this study are available from the authors on reasonable request.

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

## Acknowledgements

This work was supported by China Ministry of Science and Technology under Contract of 2016YFA (0202801), the National Natural Science Foundation of China (21521091, 21390393, U1463202, 21471089, 21671117, 21703219), and China Postdoctoral Science Foundation (2017M620738).

## Author contributions

Y.Z. conceived the idea, proposed the strategy, designed and performed the experiment, analyzed the results, and wrote the manuscript. W.S. conducted the DFT calculations. J.L. provided AC HAADF-STEM characterization. W.C., L.Z., and J.D. helped with the XAFS characterization and corresponding data analysis. T.C. and Y.H. assisted in HR-TEM, STEM, and EDX elemental mapping characterizations. J.Z. and M.Z. helped with the catalyst evaluation and corresponding data analysis. C.C. and Q.P. helped with data analyses and discussions. D.W. and Y.L. supervised the project, helped design the experiments, evaluated the data, and wrote the manuscript. The results of the manuscript were discussed by all authors.

## Additional information

**Competing interests:** The authors declare no competing interests.

