## [Peer Review File · Nature Communications]

Reviewers' comments:

Reviewer #1 (Remarks to the Author):

Prof. Li and his co-workers give us a very interesting research about the Co single site for benzene oxidation to phenol. The conversion of benzene oxidation is up to 68%, which is impressive. The characterization of the catalyst is rich and detailed. I recommend to publish this work after a minor revision.

Several comments for your information.

1. In the study, the CoN₄ center is considered as the active site. The authors may provide more evidence to show why only CoN₄ was formed during process of the thermal treating the precursor at high temperature. Are there CoN₃, CoN₂, CoN or CoC in the matrix of the carbon?
2. The author may provide more discussions about the structure of the different types of CoN_x and their effect on the catalytic performance. If the experiment is difficult, at least, the DFT calculations may provide an explanation. This literature (Angew. Chem. Int. Ed. 2016, 55, 6708–6712) is for your reference.

Reviewer #2 (Remarks to the Author):

It's very impressive that the authors using the lamella-like layers with protein β -sheets derives from cocoon silk as the precursor of porous carbon material. Single-site Co species anchored on nitrogen-doped porous carbon nanosheets (Co-ISA/CNS) were obtained during the salts promoted pyrolysis process, and the resulting catalyst exhibited remarkable catalytic activity in direct benzene oxidation. The manuscript is publishable in Nature Communications.

1. Line 46-47, in Introduction: "the supported single-site catalysts with ultrahigh surface area have been rarely reported." is misleading. Actually, zeolites, MOFs, and a lots of carbon-based materials supported single site catalysts have ultrahigh surface area. Thus, please make adjustments on the claim to make sure that it reflects the reality.
2. The authors claimed that the Co species in recycled Co-ISA/CNS was in high-spin Co(IV) state, but the EPR signal also represents the typical Co(II) species, for example, Co₃O₄. The authors should identify the valence of Co more carefully. By the way, it would be better if the author could provide the initial EPR state of the Co-ISA/CNS and the changes following the addition of H₂O₂. The EXAFS wavelet transforms analysis and XANES fitting of the used catalysts had better be provided.

3. In the computational part, “we performed comparative density functional theory calculations on the reaction mechanisms of benzene oxidation over Co-ISA/CNS, CoPc, and Co nanoparticles (Co₆ cluster).” However, the reaction mechanisms of CoPc could not be found in the manuscript and the Supplementary Materials. Fig 4a demonstrated that Co-ISA/CNS and CoPc had dramatically different reactivity towards benzene oxidation. Thus, providing a theoretical explanation on it would attract broader and more interesting!
4. The author has reported some CN_x materials supported Co single-site catalysts by other approaches, especially for the single atoms with different N coordination number, the readers would appreciate it if the authors could compare these single-atom catalysts in oxidation of benzene, and provide structure-activity relationship analysis or other insight.

Reviewer #3 (Remarks to the Author):

In this manuscript, Zhu et. al has studied cocoon silk chemistry strategy to ultrathin N-doped carbon nanosheet with metal single-site catalysts for C-H bond activation. After careful evaluation of the manuscript, I find this manuscript is not suitable for publication in Nature Communications. Detailed comments are as follows:

1. The introduction need to be upgraded with more recent references on the topic.
2. As author mentioned 68% conversion of benzene with 61% yield of phenol is the highest value reported for heterogeneous catalyst at room temperature. But according to underneath reference, benzene conversion is 68.5% with 65% yield of phenol at 80 °C which is not a harsh condition. Moreover benzene: H₂O₂ (1:12) mole ratio is very high compare to the underneath reference which is 1:5, then what is the novelty of your reaction? (ACS Appl. Mater. Interfaces ACS Appl. Mater. Interfaces, 2014, 6, 14451–14459)
3. Author should mention the actual reaction temperature of the reaction.
4. Author should mention hydrogen peroxide efficiency for this reaction.
5. What is the significance of zinc salt during synthesis? Author should mention whether it is leaching out completely or it remain intact on the surface or intercalated into layer space. XPS, EXAFS and ICP data shows only the presence of Co. If it is not present in the catalyst then why author is using zinc salt during synthesis?
6. Author should mention the selectivity of phenol as well as other side products?
7. There are no optimized temperature conditions for this reaction. As author mention the reaction pathway is exothermic and low temperature is favourable for this reaction then they should give effect of temperature on yield of phenol by increasing and decreasing temperature.
8. Author should study the effect of solvent.
9. Author also synthesized Fe-ISA/CNS and Ni-ISA/CNS catalyst but did not compare the catalyst activity and selectivity with Co-ISA/CNS catalyst.
10. Author should give XPS data of spent catalyst. And author did not explain the XPS data of

freshly prepared Co-ISA/CNS catalyst like oxidation state of cobalt atom.

Reviewer(s)' Comments to Author:

Reviewer: 1

Comments:

Prof. Li and his co-workers give us a very interesting research about the Co single site for benzene oxidation to phenol. The conversion of benzene oxidation is up to 68%, which is impressive. The characterization of the catalyst is rich and detailed. I recommend to publish this work after a minor revision.

1) In the study, the CoN_4 center is considered as the active site. The authors may provide more evidence to show why only CoN_4 was formed during process of the thermal treating the precursor at high temperature. Are there CoN_3 , CoN_2 , CoN or CoC in the matrix of the carbon?

Re: Thanks for the contributing suggestion. We have developed a cocoon silk chemistry strategy to synthesize isolated Co single-site catalysts anchored on N-doped porous carbon nanosheet. Based on the present experiments and our group's published works, as well as the DFT calculations, we find that the coordination number and surrounding environment of central Co sites can be regulated by controlling the pyrolysis temperatures. The coordination bonds between N and Co atoms in pyrolytic carbon matrix can be partly broken up with pyrolysis temperature increasing, leading to various Co-N atomic structures. At 900 °C pyrolysis temperature, the atomic structures for Co species are mainly present as CoN_4 configuration and are implicated as the dominant active sites in Co-ISA/CNS catalyst. For 950 and 1000 °C, the Co-N coordination centers mainly consist of CoN_3 and CoN_2 , respectively. More discussions about the finding and supporting information please see:

The statement "So based on the experimental spectra results and the well-matched EXAFS fitting, the local atomic structure model of Co-ISA/CNS can be identified as shown by the inset in Fig. 2e, where one Co atom is coordinated by four N atoms (CoN_4) confined in the carbon matrix. However, it is necessary to point out that the pyrolytic carbonaceous materials converted from metal-contained precursors may usually deliver various metal-nitrogen moieties (MN_x , $x = 0, 1, 2, 3$, and 4) in carbon lattices caused by the disorder, edge, and in-plane atomic vacancy during the thermal treating process at high temperature. Furthermore it is difficult to completely exclude the existence of other metal-nitrogen moieties based on the current experimental characterization technology. So we carried out the density functional theory (DFT) calculations on formation energies of various possible structures to examine the stability of CoN_x ($x = 0, 1, 2, 3$, and 4) and to access whether there are CoN_3 , CoN_2 , CoN or CoC existing in the carbon matrix. The DFT calculations were conducted following a modified procedure based on that reported in Deng's work⁴¹, and the detailed methods and results are summarized in Supplementary Fig. 9. After comparing the formation energies of different structures, it is evidently found that the CoN_4 moiety is the most stable compound among MN_x structures. Therefore we propose that the CoN_4 moieties are the most possible atomic structures for Co species and serve as the dominant active sites in Co-ISA/CNS catalyst." was added in the revised manuscript. Please see in page 12 line 3. The additional supporting data are also provided as shown in Supplementary Fig. 9, Please see in the revised SI.

Supplementary Figure S9 | (a) and (b) Calculated structural stability of Co-ISA/CNS catalysts with various N coordination numbers in different CoN_x models (x = 0, 1, 2, 3, and 4).

Reference

- Cui, X. *et al.* A graphene composite material with single cobalt active sites: a highly efficient counter electrode for dye-sensitized solar cells. *Angew. Chem. Int. Ed.* **55**, 6708–6712 (2016).

More discussions about the finding and supporting information also please see:

The statement “Furthermore, we also found that the cocoon silk chemistry strategy proves versatile in synthesis of Co single-site catalysts with tailorable central-metal atomic local structures. Our previously published researches have demonstrated that a certain amount of coordination bonds between N and metal atoms in pyrolytic carbon matrix can be broken up with pyrolysis temperature increasing⁶⁰. Herein we synthesized different samples at much higher pyrolysis temperature. Microstructure characterizations and XAFS analysis (Supplementary Fig. 11,12,13,14) suggest that both the samples exhibit a similar morphology of the ultrathin 2D porous N-doped carbon nanosheets with atomically dispersed Co sites. The dominant reactive sites for Co single-site catalysts obtained at 950 and 1000 °C are identified as CoN₃ and CoN₂, designated as Co-ISA/CNS-3 and Co-ISA/CNS-2, respectively.” was added in the revised manuscript. Please see in page 13 line 13. The additional supporting data are also provided as shown in Supplementary Fig. 11,12,13,14. Please see in the revised SI.

Supplementary Figure S11 | Chemical composition and physical characterizations of Co-ISA/CNS-3 catalyst. (a) XRD pattern, (b) Raman spectrum, (c) and (d) high-resolution N 1s and Co 2p XPS spectra.

Supplementary Figure S12 | Microstructure and morphology characterizations of Co-ISA/CNS-3 catalyst. (a) TEM image, (b) HRTEM image, (c) and (d) HAADF STEM image and corresponding EDX elemental mapping, (e) Aberration-corrected HAADF-STEM image, (f) Fourier transformed (FT) k^3 -weighted $\chi(k)$ -function of the EXAFS spectra for Co K-edge and corresponding EXAFS fitting curves at R space.

Supplementary Figure 13 | Chemical composition and physical characterizations of Co-ISA/CNS-2 catalyst. (a) XRD pattern, (b) Raman spectrum, (c) and (d) high-resolution N1s and Co 2p XPS spectra.

Supplementary Figure 14 | Microstructure and morphology characterizations of Co-ISA/CNS-2 catalyst. (a) TEM image, (b) HRTEM image, (c) and (d) HAADF STEM image and corresponding

EDX elemental mapping, (e) Aberration-corrected HAADF-STEM image, (f) Fourier transformed (FT) k^3 -weighted $\chi(k)$ -function of the EXAFS spectra for Co K-edge and corresponding EXAFS fitting curves at R space.

Reference

60. Wang, X. *et al.* Regulation of coordination number over single Co sites: triggering the efficient electroreduction of CO_2 . *Angew. Chem. Int. Ed.* **57**, 1944–1948 (2018).

2) The author may provide more discussions about the structure of the different types of CoN_x and their effect on the catalytic performance. If the experiment is difficult, at least, the DFT calculations may provide an explanation. This literature (*Angew. Chem. Int. Ed.* 2016, 55, 6708–6712) is for your reference.

Re: Thanks for the good suggestion. We have checked the catalytic performance of the Co-ISA/CNS catalyst with different types of CoN_x structures. Our experimental result reveals that the Co-ISA/CNS catalyst with CoN_4 structure achieved the highest catalytic activity. More discussions please see:

The statement “In order to insight into the structure-activity relationship of the as-synthesized Co single-site catalyst towards the benzene oxidation reaction, we also measured the catalytic performance of another two samples with different N coordination numbers under the same conditions. As revealed in Supplementary Fig. 29, the Co-ISA/CNS catalyst with CoN_4 structure achieved the highest catalytic activity in comparison with Co-ISA/CNS-3 (CoN_3) and Co-ISA/CNS-2 (CoN_2). The resultant difference in catalytic activity is speculated to be related to their structural stability in practice reaction conditions. In case of the most stable CoN_4 structure, the sufficient reactive centers can be effectively build up and maintain the longest lifetime, which could endow the highest catalytic activity. In transition-metal-catalyzed C–H bond activation, the generated metal sites play a key role in achieving desirable catalytic activity.” was added in the revised manuscript. Please see in page 18 line 11. The additional supporting data are also provided as shown in Supplementary Fig. 29. Please see in the revised SI.

Supplementary Figure 29 | The performance for the direct oxidation of benzene to phenol catalyzed by Co-ISA/CNS catalysts with different N coordination number.

Reviewer: 2

Comments:

It's very impressive that the authors using the lamella-like layers with protein β -sheets derives from cocoon silk as the precursor of porous carbon material. Single-site Co species anchored on nitrogen-doped porous carbon nanosheets (Co-ISA/CNS) were obtained during the salts promoted pyrolysis process, and the resulting catalyst exhibited remarkable catalytic activity in direct benzene oxidation. The manuscript is publishable in *Nature Communications*.

1) Line 46-47, in Introduction: "the supported single-site catalysts with ultrahigh surface area have been rarely reported." is misleading. Actually, zeolites, MOFs, and a lots of carbon-based materials supported single site catalysts have ultrahigh surface area. Thus, please make adjustments on the claim to make sure that it reflects the reality.

Re: Thanks for the useful comment. We are sorry for giving a misleading statement. Following a careful consideration, we have adjusted the sentence. The revised statement please see:

The statement "Moreover, significantly enlarging surface area of supports can expose the most active sites accessible to reactants, which has been considered as an effective route to boost the intrinsic activity of these heterogeneous supported single-site catalysts³⁰. Various zeolites, metal-organic frameworks (MOFs), and well-defined porous carbon materials have been employed as possible support for single-site catalysts³¹⁻³⁴." was added in the revised manuscript. Please see in page 3 line 13.

References

30. Bell, A. T. The impact of nanoscience on heterogeneous catalysis. *Science* **299**, 1688–1691 (2003).
31. Nozaki, C., Lugmair, C. G., Bell, A. T. & Tilley, T. D. Synthesis, characterization, and catalytic performance of single-site iron (III) centers on the surface of SBA-15 silica. *J. Am. Chem. Soc.* **124**, 13194–13203 (2002).
32. Kistler, J. D. *et al.* A single-site platinum CO oxidation catalyst in zeolite KLTL: microscopic and spectroscopic determination of the locations of the platinum atoms. *Angew. Chem. Int. Ed.* **53**, 8904–8907 (2014).
33. Li, Z. *et al.* Sintering-resistant single-site nickel catalyst supported by metal–organic framework. *J. Am. Chem. Soc.* **138**, 1977–1982 (2016).
34. Zhao, C. *et al.* Ionic exchange of metal–organic frameworks to access single nickel sites for efficient electroreduction of CO₂. *J. Am. Chem. Soc.* **139**, 8078–8081 (2017).

2) The authors claimed that the Co species in recycled Co-ISA/CNS was in high-spin Co(IV) state, but the EPR signal also represents the typical Co(II) species, for example, Co₃O₄. The authors should identify the valence of Co more carefully. By the way, it would be better if the author could provide the initial EPR state of the Co-ISA/CNS and the changes following the addition of H₂O₂. The EXAFS wavelet transforms analysis and XANES fitting of the used catalysts had better be provided.

Re: Thanks for the good suggestion. We have carefully identified the valence change of Co species in the Co-ISA/CNS catalysts under different treatment. More discussions about the findings and additional supporting data please see:

The statement "Previous computational and experimental studies have evidently revealed that the high-valent metal-oxo species generally serve as reactive intermediates in the catalytic oxidation of organic substrates⁶⁵. In particular, for the cobalt-mediated C-H oxidation reactions, the reactive intermediates are frequently implicated to cobalt(IV)-oxo species^{66,67}. However, it is difficult to identify the proposed cobalt(IV)-oxo intermediates under catalytic conditions, and only extremely rare cases have achieved some direct spectroscopic characterizations. To investigate both coordination environment and oxidation state of cobalt sites in the used Co-ISA/CNS catalysts, the cobalt K-edge XAFS measurements were first performed. As shown in Supplementary Fig. 32, the edge position was found to locate at 7721.64 eV at $A = 0.5$ edge-jump, which is ~ 0.84 eV blue-shifted from the reported Co(III)-peroxo complexes⁶⁵. Although the observed energy shift is smaller than that valence change from Co(III) to Co(IV), it is still indicative of highly oxidized cobalt sites. Additionally, the used Co-ISA/CNS catalysts delivered higher white line intensity than that of the fresh Co-ISA/CNS catalysts, demonstrating the more positive average oxidation state of cobalt sites in the used Co-ISA/CNS catalysts²⁸. The EXAFS curves of the used Co-ISA/CNS catalysts were best modelled with cobalt sites in an about five number N/O coordination environment. These XAFS results strongly indicate that the cobalt active sites in the Co-ISA/CNS catalysts are generated as a square pyramidal $\text{Co}^{\text{IV}}\text{-O}$ -like species⁶⁵. This conclusion is also well evidenced by the electron paramagnetic resonance (EPR) spectroscopy and XPS data analysis of the Co-ISA/CNS samples before and after the H_2O_2 treatment. As shown in Supplementary Fig. 34a, EPR spectrum of the Co-ISA/CNS catalyst with H_2O_2 treatment shows signals at $g = 6.6$ and 2.01, indicating the spin-quartet ($S = 3/2$) ground state^{65,68}. The enhanced EPR signals could be ascribed to the probable formation of Co=O/O=Co=O , leading to Co 3d mixing with O 2p. The interaction between Co atomic sites and activated oxygen species will enhance the unoccupied state of Co species because of the electronegativity of cobalt-bonding O atoms. Supplementary Fig. 34b presents the binding energy variety of Co 2p from the fresh, H_2O_2 treated and spent Co-ISA/CNS catalysts based on XPS analysis. All spectra exhibit one pair of peaks relative to Co $2p_{3/2}$ and Co $2p_{1/2}$. The Co $2p_{3/2}$ of the fresh Co-ISA/CNS catalysts occurs at 780.8 eV, which is slightly higher than that of Co^{II} but smaller than Co^{III} chemical state. Interestingly, the Co $2p_{3/2}$ displays a positive energy shift of ~ 1.1 eV after H_2O_2 treatment, which is higher than the value of Co^{III} chemical state in literature⁶⁹. Thus, our current findings should support the involvement of the O=Co=O center as the most plausible reactive intermediate in this cobalt-catalyzed oxidation reaction." was added in the revised manuscript. Please see in page 20 line 13. The additional supporting data are also provided as shown in Supplementary Fig. 32 and 34. Please see in the revised SI.

Supplementary Figure 32 | XAFS measurement of recycled Co-ISA/CNS catalyst. (a) Co K-edge XANES spectra of the recycled Co-ISA/CNS and fresh Co-ISA/CNS. (b) Fourier transformed (FT) k^3 -weighted $\chi(k)$ -function of the EXAFS spectra for Co K-edge. (c) and (d) EXAFS curves at k space and R space, respectively.

Supplementary Figure 34 | (a) EPR spectra of Co-ISA/CNS catalyst with H_2O_2 treatment (red line) in comparison to the corresponding original sample (black line). **(b)** High-resolution Co 2p XPS spectra of Co-ISA/CNS catalyst before and after H_2O_2 treatment.

References

28. Chen, W. *et al.* Single tungsten atoms supported on MOF-derived N-doped carbon for robust electrochemical hydrogen evolution. *Adv. Mater.* DOI: 10.1002/adma.201800396 (2018).
65. Wang, B. *et al.* Synthesis and reactivity of a mononuclear non-haem cobalt (IV)-oxo complex. *Nat. Commun.* **8**, 14839 (2017).
66. Nguyen, A. I. *et al.* Mechanistic investigations of water oxidation by a molecular cobalt oxide analogue: evidence for a highly oxidized intermediate and exclusive terminal oxo participation. *J. Am. Chem. Soc.* **137**, 12865–12872 (2015).
67. Hadt, R. G. *et al.* X-ray spectroscopic characterization of Co(IV) and metalmetal interactions in Co_4O_4 : Electronic structure contributions to the formation of high-valent states relevant to the oxygen evolution reaction. *J. Am. Chem. Soc.* **138**, 11017–11030 (2016).
68. Hong, S. *et al.* Spectroscopic capture and reactivity of a low-spin cobalt (IV)-oxo complex stabilized by binding redox-inactive metal ions. *Angew. Chem. Int. Ed.* **53**, 10403–10407 (2014).

69. Liao, L. *et al.* Efficient solar water-splitting using a nanocrystalline CoO photocatalyst. *Nat. Nanotech.* **9**, 69–73 (2014).

3) In the computational part, “we performed comparative density functional theory calculations on the reaction mechanisms of benzene oxidation over Co-ISA/CNS, CoPc, and Co nanoparticles (Co₆ cluster).” However, the reaction mechanisms of CoPc could not be found in the manuscript and the Supplementary Materials. Fig 4a demonstrated that Co-ISA/CNS and CoPc had dramatically different reactivity towards benzene oxidation. Thus, providing a theoretical explanation on it would attract broader and more interesting!

Re: Thanks for the good comment. We have provided theoretical explanation and discussions about the findings and additional supporting data please see:

A statement “Supplementary Fig. 36 shows that benzene oxidation over CoPc undergoes a similar reaction pathway to that on Co-ISA/CNS, but the H₂O₂ molecule was initially dissociated on Co site forming an OH-Co-OH intermediate. In the H₂O₂-H₂O-CH₃CN system, the OH-Co-OH intermediate was easily destroyed for the observation of severely bubbling in the practical oxidation reaction, indicative of a strong tendency to self-decomposition of hydrogen peroxide rather than contributing to hydroxylation reaction, which may break off the following processes and result into the deactivation of Co site.” was added in the revised manuscript. Please see in page 22 line 8. The additional supporting data are also provided as shown in Supplementary Fig. 36. Please see in the revised SI.

Supplementary Figure 36 | Free energy diagram of the oxidation of benzene to phenol on CoPc. The gray, blue, red, and white balls represent C, Co, O, and H atoms, respectively.

4) The author has reported some CN_x materials supported Co single-site catalysts by other approaches, especially for the single atoms with different N coordination number, the readers would appreciate it if the authors could compare these single-atom catalysts in oxidation of benzene, and provide structure-activity relationship analysis or other insight.

Re: Thanks for the good comment. Through controlling the pyrolysis temperature, we synthesized three Co-ISA/CNS catalysts with different types of CoN_x structures and compared their catalytic performance. Our experimental result reveals that the Co-ISA/CNS catalyst with CoN_4 structure achieved the highest catalytic activity. More discussions please see:

A statement “In order to insight into the structure-activity relationship of the as-synthesized Co single-site catalyst towards the benzene oxidation reaction, we also measured the catalytic performance of another two samples with different N coordination numbers under the same conditions. As revealed in Supplementary Fig. 29, the Co-ISA/CNS catalyst with CoN_4 structure achieved the highest catalytic activity in comparison with Co-ISA/CNS-3 (CoN_3) and Co-ISA/CNS-2 (CoN_2). The resultant difference in catalytic activity is speculated to be related to their structural stability in practice reaction conditions. In case of the most stable CoN_4 structure, the sufficient reactive centers can be effectively build up and maintain the longest lifetime, which could endow the highest catalytic activity. In transition-metal-catalyzed C–H bond activation, the generated metal sites play a key role in achieving desirable catalytic activity.” was added in the revised manuscript. Please see in page 18 line 11. The additional supporting data are also provided as shown in Supplementary Fig. 29. Please see in the revised SI.

Supplementary Figure 29 | The performance for the direct oxidation of benzene to phenol catalyzed by Co-ISA/CNS catalysts with different N coordination number.

Reviewer: 3

Comments:

In this manuscript, Zhu *et. al* has studied a cocoon silk chemistry strategy to ultrathin N-doped carbon nanosheet with metal single-site catalysts for C-H bond activation. After careful evaluation of the manuscript, I find this manuscript is not suitable for publication in *Nature Communications*. Detailed comments are as follows:

1) The introduction need to be upgraded with more recent references on the topic.

Re: Thanks for the useful comment. We have adjusted the introduction accordingly. The statement about the topic of direct catalytic oxidation of benzene to phenol with hydrogen peroxide was moved from the result part to the introduction. Lots of recent references were also added. More discussions about the topic and additional supporting data please see:

The statement (1) "Indeed, single-site catalysts, including Au^{7,8}, Pt^{9,10}, Pd^{11,12}, Ru^{13,14}, Ir^{15,16}, Rh¹⁷, Fe¹⁸⁻²⁰, Co^{21,22}, Ni²³⁻²⁵, Mn²⁶, Mo²⁷, and W²⁸, are reported....." and

The statement (2) "Moreover, significantly enlarging surface area of supports can expose the most active sites accessible to reactants, which has been considered as an effective route to boost the intrinsic activity of these heterogeneous supported single-site catalysts³⁰. Various zeolites, metal-organic frameworks (MOFs), and well-defined porous carbon materials have been employed as possible support for single-site catalysts³¹⁻³⁴." and

The statement (3) "Considerable researches have demonstrated that support materials can not only firmly stabilize isolated single metal sites but often play a key role in controlling the local geometric and electronic structures of catalytic sites, leading to excellent retentivity of individual site state under catalytic conditions and superb catalytic activity^{1,14,37}. To date, the reported support materials generally consist of the oxides and carbon-based materials³⁸." and

The statement (4) "The selective catalytic oxidation of aromatic inactive C-H bonds to chemical intermediates is one of the most active research topics but still remains a long-standing challenge⁴³⁻⁴⁶. Phenol is the most important industrial chemical due to its broad application in the fields of phenolic resin, pharmaceuticals, agrochemicals, etc^{47,48}. However, current phenol production is still limited to the three-step cumene process, including propylation of benzene, autoxidation to cumene hydroperoxide, and Hock rearrangement⁴⁹⁻⁵². These reactions are conducted under elevated temperature, high pressure, and strongly acidic conditions, which result a low overall yield of phenol from benzene (< 5%) and an equimolar amount of acetone byproduct^{50,51}. Thus, one-step direct catalytic oxidation of benzene to phenol with hydrogen peroxide over heterogeneous catalysts has remained a focal subject of the fundamental and applied research^{4,18,53-55}. Bal and co-workers have devoted great effort in such catalytic system based on novel heterogeneous catalysts including zeolite and CuCr₂O₄ spinel^{43-45,48,56}. Despite considerable achievements, the direct catalytic oxidation of benzene at room temperature is still very difficult due to the notoriously low reactivity of the aromatic C-H bonds^{57,58}. Development of more efficient catalysts has been the most urgent task. Given that the achieved high catalytic performance in most single-site catalysts, we expect their preparation will provide exciting opportunity in direct catalytic oxidation of benzene to phenol." were added in introduction in the revised manuscript. Please see in page 3 and page 4. The year distribution of the cited references

is shown as in following figure. Please see:

Supplementary Figure of the year distribution of the cited references in whole revised manuscript.

Upgraded references in introduction please see:

References (1)

- Chen, Z. *et al.* Single-site Au^I catalyst for silane oxidation with water. *Adv. Mater.* **30**, 1704720 (2018).
- Chen, Y. *et al.* Discovering partially charged single-atom Pt for enhanced anti-markovnikov alkene hydrosilylation. *J. Am. Chem. Soc.* **140**, 7407–7410 (2018).
- Chen, Z. *et al.* A heterogeneous single-atom palladium catalyst surpassing homogeneous systems for Suzuki coupling. *Nat. Nanotech.* DOI: 10.1038/s41565-018-0167-2 (2018).
- Yang, J. *et al.* Phosphorus nitride imide nanotube as carbon-free support to anchor single Ru sites for efficient and robust hydrogen evolution. *Angew. Chem. Int. Ed.* DOI: 10.1002/anie.201804854 (2018).
- Zhang, Y. *et al.* Atomic iridium incorporated in cobalt hydroxide for efficient oxygen evolution catalysis in neutral electrolyte. *Adv. Mater.* **30**, 1707522 (2018).
- Zhang, Z., Sun, J., Wang, F. & Dai, L. Efficient oxygen reduction reaction (ORR) catalysts based on single iron atoms dispersed on a hierarchically structured porous carbon framework. *Angew. Chem. Int. Ed.* DOI: 10.1002/anie.201804958 (2018).
- Xue, Y. *et al.* Anchoring zero valence single atoms of nickel and iron on graphdiyne for hydrogen evolution. *Nat. Commun.* **9**, 1460 (2018).
- Yang, Y. *et al.* O-, N-atoms-coordinated Mn cofactors within a graphene framework as bioinspired oxygen reduction reaction electrocatalysts. *Adv. Mater.* DOI: 10.1002/adma.201801732 (2018).
- Chen, W. *et al.* Rational design of single molybdenum atoms anchored on N-doped carbon for effective hydrogen evolution reaction. *Angew. Chem. Int. Ed.* **56**, 16086–16090 (2017).
- Chen, W. *et al.* Single tungsten atoms supported on MOF-derived N-doped carbon for robust electrochemical hydrogen evolution. *Adv. Mater.* DOI: 10.1002/adma.201800396 (2018).

References (2)

- Bell, A. T. The impact of nanoscience on heterogeneous catalysis. *Science* **299**, 1688–1691

(2003).

31. Nozaki, C., Lugmair, C. G., Bell, A. T. & Tilley, T. D. Synthesis, characterization, and catalytic performance of single-site iron (III) centers on the surface of SBA-15 silica. *J. Am. Chem. Soc.* **124**, 13194–13203 (2002).
32. Kistler, J. D. *et al.* A single-site platinum CO oxidation catalyst in zeolite KLTL: microscopic and spectroscopic determination of the locations of the platinum atoms. *Angew. Chem. Int. Ed.* **53**, 8904–8907 (2014).
33. Li, Z. *et al.* Sintering-resistant single-site nickel catalyst supported by metal–organic framework. *J. Am. Chem. Soc.* **138**, 1977–1982 (2016).
34. Zhao, C. *et al.* Ionic exchange of metal–organic frameworks to access single nickel sites for efficient electroreduction of CO₂. *J. Am. Chem. Soc.* **139**, 8078–8081 (2017).

References (3)

14. Yang, J. *et al.* Phosphorus nitride imide nanotube as carbon-free support to anchor single Ru sites for efficient and robust hydrogen evolution. *Angew. Chem. Int. Ed.* DOI: 10.1002/anie.201804854 (2018).
37. Zhu, C., Fu, S., Shi, Q., Du, D. & Lin, Y. Single-atom electrocatalysts. *Angew. Chem. Int. Ed.* **56**, 13944–13960 (2017).
38. Wang, A., Li, J. & Zhang, T. Heterogeneous single-atom catalysis. *Nat. Rev. Chem.* **2**, 65–81 (2018).

References (4)

43. Acharyya, S. S., Ghosh, S., Adak, S., Sasaki, T. & Bal, R. Facile synthesis of CuCr₂O₄ spinel nanoparticles: a recyclable heterogeneous catalyst for the one pot hydroxylation of benzene. *Catal. Sci. Technol.* **4**, 4232–4241 (2014).
44. Acharyya, S. S. *et al.* Synergistic effect between ultrasmall Cu(II) oxide and CuCr₂O₄ spinel nanoparticles in selective hydroxylation of benzene to phenol with air as oxidant. *ACS Catal.* **5**, 2850–2858 (2015).
45. Acharyya, S. S., Ghosh, S., Siddiqui, N., Konathala, L. S. & Bal, R. Cetyl alcohol mediated synthesis of CuCr₂O₄ spinel nanoparticles: a green catalyst for selective oxidation of aromatic C–H bonds with hydrogen peroxide. *RSC Adv.* **5**, 4838–4843 (2015).
46. Ramu, R. *et al.* Mechanistic study for the selective oxidation of benzene and toluene catalyzed by Fe(ClO₄)₂ in an H₂O₂–H₂O–CH₃CN system. *Mol. Catal.* **441**, 114–121 (2017).
47. Wei, Q., Fan, H., Qin, F., Ma, Q. & Shen, W. Metal-free honeycomb-like porous carbon as catalyst for direct oxidation of benzene to phenol. *Carbon*, **133**, 6–13 (2018).
48. Bal, R., Tada, M., Sasaki, T. & Iwasawa, Y. Direct phenol synthesis by selective oxidation of benzene with molecular oxygen on an interstitial-N/Re cluster/zeolite catalyst. *Angew. Chem. Int. Ed.* **45**, 448–452 (2006).
56. Acharyya, S. S., Ghosh, S. & Bal, R. Fabrication of three-dimensional (3D) raspberry-like copper chromite spinel catalyst in a facile hydrothermal route and its activity in selective hydroxylation of benzene to phenol. *ACS Appl. Mater. Interfaces*, **6**, 14451–14459 (2014).

2) As author mentioned 68% conversion of benzene with 61% yield of phenol is the highest value reported for heterogeneous catalyst at room temperature. But according to underneath reference, benzene conversion is 68.5% with 65% yield of phenol at 80 °C which is not a harsh condition. Moreover benzene: H₂O₂ (1:12) mole ratio is very high compare to the underneath

reference which is 1:5, then what is the novelty of your reaction? (ACS Appl. Mater. Interfaces ACS Appl. Mater. Interfaces, 2014, 6, 14451-14459)

Re: Thanks for the useful comment. The novelty of our Co-ISA/CNS-catalyzed reaction is relative to the formation of the O=Co=O active center and thus the novel reaction pathway for the direct catalytic oxidation of benzene to phenol with hydrogen peroxide at room temperature. Over our Co-ISA/CNS catalyst, one H₂O₂ molecule is first adsorbed on the single Co atomic site and then dissociated to form a Co=O intermediate with releasing one water molecule. After that, another H₂O₂ molecule was continuously adsorbed and dissociated on the other side of the Co atomic site, and then an O=Co=O center was established. The generated O=Co=O center could present activated oxygen species to adsorb the benzene molecule via the C–O bonding. The adsorbed benzene was subsequently converted to phenol via an H atom transfer from adjacent C to O. DFT calculations reveal that reaction pathway is exothermic and favorable for low-temperature reactions. We find that the Co-single-sites-catalyzed reactions experience a different pathway with that of the homogeneous metal Co nanoparticle and transition metal oxide catalysts. For transition metal oxide catalysts, such as CuCr₂O₄ spinel, the H₂O₂ is adsorbed on catalyst surface and dissociates into free hydroxy radicals (•OH). Subsequently, the generated hydroxy radicals behave as active species to attack the activated C–H bond of benzene, leading to homolytic C–H bond cleavage to generate hydroxylated benzene over catalyst surface. So we believe that our Co-ISA/CNS-catalyzed reaction should provide a novelty in reaction pathway much more superior to that over traditional heterogeneous catalysts. More discussions about the reaction mechanism please see the detailed statement in DFT calculation analysis:

The statement “Therefore, the high activity for benzene oxidation to phenol over Co-ISA/CNS catalyst could be ascribed to the intrinsic coordination environment and the atomic dispersion of Co species in Co-ISA/CNS, which provides a novelty in reaction pathway much more superior to that over traditional heterogeneous catalysts.” was added in the revised manuscript. Please see in page 22 and page 17.

3) Author should mention the actual reaction temperature of the reaction.

Re: Thanks for the useful comment. We have defined the room temperature of the benzene oxidation reaction as actual value of 25 °C through the whole revised manuscript. Please see:

The statement “The reaction was conducted in a 25 ml sealed glass reactor with 50 mg catalyst, 0.4 ml benzene, 6 ml H₂O₂ (30%), and 3 ml CH₃CN at room temperature (25 °C).” was added in the revised manuscript. Please see in page 24 line 9.

4) Author should mention hydrogen peroxide efficiency for this reaction.

Re: Thanks for the useful comment. We have provided hydrogen peroxide efficiency of the benzene oxidation reaction in the revised manuscript. Please see:

The statement “Additionally, as shown in Supplementary Fig. 25 that the hydrogen peroxide efficiency over the Co-ISA/CNS catalyst increased from 1.1% to 4.7% in the catalytic reaction process, which is related to the H₂O₂ decomposition rate and the benzene oxidation rate.” was added in the revised manuscript. Please see in page 17 line 12. The additional supporting data is

also provided as shown in Supplementary Fig. 25. Please see in the revised SI.

Supplementary Figure 25 | The H₂O₂ efficiency in the direct oxidation of benzene to phenol catalyzed by Co-ISA/CNS catalyst with different reaction times. Reaction conditions: 50 mg catalyst, 0.4 ml benzene, 6 ml H₂O₂ (30%), and 3 ml CH₃CN at room temperature (25 °C) for different reaction times. H₂O₂ efficiency = [moles of phenol formed/total moles of H₂O₂ added] × 100.

5) What is the significance of zinc salt during synthesis? Author should mention whether it is leaching out completely or it remain intact on the surface or intercalated into layer space. XPS, EXAFS and ICP data shows only the presence of Co. If it is not present in the catalyst then why author is using zinc salt during synthesis?

Re: Thanks for the useful comment. We used the zinc salt (ZnCl₂) as the chemical activating reagent in the pyrolysis to produce porous carbon materials with ultrahigh specific surface area. At elevated temperature beyond the boiling point of ZnCl₂ (756 °C), the zinc salt can be evaporated away completely, where etching effect toward C–C bonds occurs, leaving numerous carbon atom vacancy and further evolving into abundant nanopores. More discussions about the topic and supporting data please see:

The statement “Additionally, the present cocoon silk chemistry strategy is also versatile in synthesis of porous carbon materials with ultrahigh specific surface area. In general, high surface areas and large pore volumes are highly required for heterogeneous catalysts. Chemical activation followed by carbonization process has been extensively employed to achieve such materials using different chemical activating reagents, typically KOH, H₃PO₄ or ZnCl₂, etc^{61,62}. The ZnCl₂ activation approach can enjoy many advantages of low cost, and minor erosion. During carbonization process, the ZnCl₂ salt shows strong interaction or coordination with the carbon π-electron organism. At elevated temperature beyond the boiling point of ZnCl₂ (756 °C)⁶², it can be evaporated away completely, where etching effect toward C–C bonds takes place, leaving numerous carbon atom vacancy and further evolving into abundant nanopores. So it can be concluded that the quantity of ZnCl₂ salt used in synthesis has crucial impacts upon the as-synthesized carbon texture. Supplementary Fig. 15,16 show the microstructure characterizations of the Co-ISA/CNS catalysts synthesized with reduced amount of with ZnCl₂ salt

(5.5 mmol) and in the absence of ZnCl_2 salt, respectively. Although the typical ultrathin 2D morphology can be well maintained, both the samples exhibit much less nanopores. Especially, the Co-ISA/CNS catalysts synthesized without ZnCl_2 salt show a relatively smooth surface, indicative of disappearing of in-plane porous structure. The BET surface areas of Co-ISA/CNS catalysts synthesized with 5.5 mmol and without ZnCl_2 salt are measured to be 1781 and 1061 $\text{m}^2 \text{g}^{-1}$ (Supplementary Fig. 17).” was added in the revised manuscript. Please see in page 14 line 10. The additional supporting data are also provided as shown in Supplementary Fig. 15,16 and 17. Please see in the revised SI.

Supplementary Figure 15 | Microstructure and morphology characterizations of Co-ISA/CNS catalyst synthesized with 5.5 mmol ZnCl_2 . (a) Typical TEM image, (b) HRTEM image, (c) HAADF-STEM image and (d) corresponding EDX elemental mapping.

Supplementary Figure 16 | Microstructure and morphology characterizations of Co-ISA/CNS catalyst synthesized without ZnCl₂. (a) Typical TEM image, (b) HRTEM image, (c) HAADF-STEM image and (d) corresponding EDX elemental mapping.

Supplementary Figure 17 | (a), (b), and (c) Nitrogen adsorption-desorption isotherms and the corresponding pore size distribution of Co-ISA/CNS catalyst synthesized with 5.5 mmol ZnCl₂, **(d), (e), and (f)** Nitrogen adsorption-desorption isotherms and the corresponding pore size distribution of Co-ISA/CNS catalyst synthesized without ZnCl₂.

References

61. Yu, Z. L. *et al.* Polymerization under hypersaline conditions: a robust route to phenolic polymer-derived carbon aerogels. *Angew. Chem. Int. Ed.* **55**, 14623–14627 (2016).
62. Zhang, Z. J., Dong, C., Ding, X. Y. & Xia, Y. K. A generalized ZnCl₂ activation method to produce nitrogen-containing nanoporous carbon materials for supercapacitor applications. *J. Alloys Compd.* **636**, 275–281 (2015).

6) Author should mention the selectivity of phenol as well as other side products?

Re: Thanks for the useful comment. We have provided the phenol selectivity of the benzene oxidation reaction in the revised manuscript. Please see:

The statement “Moreover, the phenol selectivity over the Co-ISA/CNS catalyst reached 97% within the initial 1 h, but decreased progressively with increasing the reaction time and maintained at a value of 90% at the reaction time prolonged to 96 h. The side-product was detected to be hydroquinone, which was over-oxidized from benzene through multiple oxidations during the liquid phase reaction process and similar to that reported in literatures^{63,64}.” was added in the revised manuscript. Please see in page 17 line 7. The additional supporting data is also provided as shown in Supplementary Fig. 24. Please see in the revised SI.

Supplementary Figure 24 | The performance for the direct oxidation of benzene to phenol catalyzed by Co-ISA/CNS catalyst with different reaction times. Reaction conditions: 50 mg catalyst, 0.4 ml benzene, 6 ml H₂O₂ (30%), and 3 ml CH₃CN at room temperature (25 °C) for different reaction times.

References

63. Yang, J. H. *et al.* Direct catalytic oxidation of benzene to phenol over metal-free graphene-based catalyst. *Energy Environ. Sci.* **6**, 793–798 (2013).
64. Balducci, L. *et al.* Direct oxidation of benzene to phenol with hydrogen peroxide over a modified titanium silicalite. *Angew. Chem. Int. Ed.* **115**, 5087–5090 (2003).

7) There are no optimized temperature conditions for this reaction. As author mention the reaction pathway is exothermic and low temperature is favourable for this reaction then they should give effect of temperature on yield of phenol by increasing and decreasing temperature.

Re: Thanks for the useful comment. We have provided the measurements about the effect of reaction temperature in benzene oxidation catalyzed by the Co-ISA/CNS catalyst. More discussions about the topic and supporting data please see:

The statement “Bal’s work demonstrated that several reaction parameters, such as reaction temperature, composition of reaction medium, amount of oxidant and type of catalyst, may have important effect on the benzene oxidation reaction in liquid phase over heterogeneous catalysts⁵⁶. Supplementary Fig. 26 suggests that the benzene oxidation reaction over the Co-ISA/CNS catalyst is highly dependent on reaction temperature. When the reaction carried out at 0°C, the Co-ISA/CNS catalyst can deliver 14.3 % benzene conversion after 24 hours, which is higher than that 8.3% of Fe-based catalyst under the identical conditions¹⁸, indicative of excellent catalytic performance. With increasing reaction temperature, both benzene conversion and phenol yields increased, but phenol selectivity decreased slightly.” was added in the revised manuscript. Please see in page 17 line 17. The additional supporting data is also provided as shown in Supplementary Fig. 26. Please see in the revised SI.

Supplementary Figure 26 | Effect of reaction temperature in benzene oxidation catalyzed by the Co-ISA/CNS catalyst. Reaction conditions: 50 mg catalyst, 0.4 ml benzene, 6 ml H₂O₂ (30%), and 3 ml CH₃CN at different temperature for 24 h.

References

18. Deng, D. *et al.* A single iron site confined in a graphene matrix for the catalytic oxidation of benzene at room temperature. *Sci. Adv.* **1**, e1500462 (2015).
56. Acharyya, S. S., Ghosh, S. & Bal, R. Fabrication of three-dimensional (3D) raspberry-like copper chromite spinel catalyst in a facile hydrothermal route and its activity in selective hydroxylation of benzene to phenol. *ACS Appl. Mater. Interfaces*, **6**, 14451–14459 (2014).

8) Author should study the effect of solvent.

Re: Thanks for the useful comment. We have provided the measurements about the effect of solvent in benzene oxidation catalyzed by the Co-ISA/CNS catalyst. More discussions about the topic and supporting data please see:

The statement “Supplementary Fig. 28 shows that solvents can also play a very critical role in benzene oxidation over the Co-ISA/CNS catalyst. Among the solvents, the acetonitrile proves to be the best reaction medium mainly due to its excellent capability to well dissolve substrate benzene and offer a highly dispersed system for solid catalyst. When reacted in the highly hydrophobic *n*-octane, the Co-ISA/CNS catalyst showed the worse activity.” was added in the revised manuscript. Please see in page 18 line 7. The additional supporting data is also provided as shown in Supplementary Fig. 28. Please see in the revised SI.

Supplementary Figure 28 | Effect of solvents in benzene oxidation catalyzed by the Co-ISA/CNS catalyst. Reaction conditions: 50 mg catalyst, 0.4 ml benzene, 6 ml H₂O₂ (30%), and 3 ml solvents at room temperature (25 °C) for 96 h.

9) Author also synthesized Fe-ISA/CNS and Ni-ISA/CNS catalyst but did not compare the catalyst activity and selectivity with Co-ISA/CNS catalyst.

Re: Thanks for the useful comment. We have presented a comparison about the catalyst activity and selectivity of the Fe-ISA/CNS, Ni-ISA/CNS and Co-ISA/CNS catalysts under the same reaction conditions. More discussions about the topic and supporting data please see:

The statement “In transition-metal-catalyzed C–H bond activation, the generated metal sites play a key role in achieving desirable catalytic activity. To clarify the effect of metal sites towards catalytic activity, Fe-ISA/CNS and Ni-ISA/CNS catalysts were also used for the benzene oxidation reaction. As shown in Supplementary Fig. 30, Fe-ISA/CNS and Ni-ISA/CNS catalysts are found to be much less active than Co-ISA/CNS catalyst, indicative of the positive role of Co sites.” was added in the revised manuscript. Please see in page 18 line 20. The additional supporting data is also provided as shown in Supplementary Fig. 30. Please see in the revised SI.

Supplementary Figure 30 | The performance for the direct oxidation of benzene to phenol catalyzed by Fe-ISA/CNS Co-ISA/CNS Ni-ISA/CNS catalysts. Reaction conditions: 50 mg catalyst, 0.4

ml benzene, 6 ml H₂O₂ (30%), and 3 ml CH₃CN at room temperature (25 °C) for 24 h.

10) Author should give XPS data of spent catalyst. And author did not explain the XPS data of freshly prepared Co-ISA/CNS catalyst like oxidation state of cobalt atom.

Re: Thanks for the useful comment. We have carefully identified the valence change of Co species in the Co-ISA/CNS catalysts under different treatment. More discussions about the findings and additional supporting data please see:

The statement "Supplementary Fig. 34b presents the binding energy variety of Co 2p from the fresh, H₂O₂ treated and spent Co-ISA/CNS catalysts based on XPS analysis. All spectra exhibit one pair of peaks relative to Co 2p_{3/2} and Co 2p_{1/2}. The Co 2p_{3/2} of the fresh Co-ISA/CNS catalysts occurs at 780.8 eV, which is slightly higher than that of Co^{II} but smaller than Co^{III} chemical state. Interestingly, the Co 2p_{3/2} displays a positive energy shift of ~1.1 eV after H₂O₂ treatment, which is higher than the value of Co^{III} chemical state in literature⁶⁹." was added in the revised manuscript. Please see in page 21 line 15. The additional supporting data is also provided as shown in Supplementary Fig. 34b. Please see in the revised SI.

Supplementary Figure 34 | (b) High-resolution Co 2p XPS spectra of Co-ISA/CNS catalyst before and after H₂O₂ treatment.

Reference

69. Liao, L. *et al.* Efficient solar water-splitting using a nanocrystalline CoO photocatalyst. *Nat. Nanotech.* **9**, 69–73 (2014).

Reviewers' Comments:

Reviewer #1 (Remarks to the Author):

Li and his co-workers give a nice revision and I would like to recommend to accept this paper on Nature communications.

Reviewer #2 (Remarks to the Author):

The authors responded to the questions in a satisfactory way. Therefore, I strongly recommend it to be published in Nat. Comm.